# Chalcone-1-Deoxynojirimycin Heterozygote Reduced the Blood Glucose Concentration and Alleviated the Adverse Symptoms and Intestinal Flora Disorder of Diabetes Mellitus Rats

**DOI:** 10.3390/molecules27217583

**Published:** 2022-11-04

**Authors:** Pin-Jian Xiao, Jia-Cheng Zeng, Ping Lin, Dao-Bang Tang, En Yuan, Yong-Gang Tu, Qing-Feng Zhang, Ji-Guang Chen, Da-Yong Peng, Zhong-Ping Yin

**Affiliations:** 1Jiangxi Key Laboratory of Natural Products and Functional Food, College of Food Science and Engineering, Jiangxi Agricultural University, Nanchang 330045, China; 2Sericultural & Agri-Food Research Institute, Guangdong Academy of Agricultural Sciences/Key Laboratory of Functional Foods, Guangdong Key Laboratory of Agricultural Products Processing, Guangzhou 510610, China; 3College of Pharmacy, Jiangxi University of Traditional Chinese Medicine, Nanchang 330004, China

**Keywords:** chalcone 1-deoxynojirimycin heterozygote, postprandial blood glucose, gut microbiota, blood biochemical analysis

## Abstract

Chalcone-1-deoxynojirimycin heterozygote (DC-5), a novel compound which was designed and synthesized in our laboratory for diabetes treatment, showed an extremely strong in vitro inhibitory activity on α-glucosidase in our previous studies. In the current research, its potential in vivo anti-diabetic effects were further investigated by integration detection and the analysis of blood glucose concentration, blood biochemical parameters, tissue section and gut microbiota of the diabetic rats. The results indicated that oral administration of DC-5 significantly reduced the fasting blood glucose and postprandial blood glucose, both in diabetic and normal rats; meanwhile, it alleviated the adverse symptoms of elevated blood lipid level and lipid metabolism disorder in diabetic rats. Furthermore, DC-5 effectively decreased the organ coefficient and alleviated the pathological changes of the liver, kidney and small intestine of the diabetic rats at the same time. Moreover, the results of 16S rDNA gene sequencing analysis suggested that DC-5 significantly increased the ratio of Firmicutes to Bacteroidetes and improved the disorder of gut microbiota in diabetic rats. In conclusion, DC-5 displayed a good therapeutic effect on the diabetic rats, and therefore had a good application prospect in hypoglycemic drugs and foods.

## 1. Introduction

Diabetes is one of the most common chronic metabolic diseases, characterized by glucose and lipid metabolism disorder and insulin dysfunction, which eventually does harm to the homeostasis in the body, including hyperlipidemia and hyperglycemia [1]. From the clinical aspect, diabetes is usually divided into two categories of type 1 diabetes (T1D) and type 2 diabetes (T2D). It is reported that T2D accounted for about 90% of all diabetic cases [2,3], and the number of diabetic patients in the world will reach 640 million by 2040 [4], which will result in serious burden on the global medical systems and human health [5].

The first-phase insulin secretion of type 2 diabetes patients is severely weakened or absent, which leads to a continuous increase in postprandial blood glucose (PBG) during most of the day [6]. Postprandial hyperglycemia is a major risk factor for diabetes complications, such as microvascular and macrovascular complications associated with diabetes [7]. Therefore, controlling postprandial blood glucose level is essential for the diabetes treatment and diabetes complications reduction [8].

In recent years, more and more pharmaceuticals have been developed for the treatment of diabetes and its complications, such as sodium-glucose cotransporter 2 (SGLT2) inhibitors [9], glucagon-like peptide-1 (GLP-1) receptor agonists [10], dipeptidyl peptidase-4 (DPP-4) inhibitors [11], advanced glycation end-products (AGE) inhibitors, protein kinase C (PKC) inhibitors [12], endothelin (ET) receptor antagonist [13], vitamin D receptor (VDR) activator [14] and α-glucosidase inhibitors (AGIs) [15,16], etc. Among them, the oral administration of AGIs has been recognized as an ideal way to control type 2 diabetes [17,18].

AGIs can effectively delay the digestion of carbohydrates through inhibiting the activity of α-glucosidase, and therefore can reduce the symptoms of post-prandial hyperglycemia and be commonly used in diabetes and its complications treatment [19]. According to different sources, AGIs are mainly divided into three categories of microbial metabolites, natural products and synthetic chemicals [20,21]. Among the various AGIs, acarbose, voglibose and miglitol show a good effect on controlling postprandial blood glucose, and has now become the most commonly used hypoglycemic pharmaceuticals [22]. However, these AGIs also bring about some adverse symptoms in the meantime, for example hyperbowel sound, abdominal distention, diarrhea, abdominal pain, nausea and vomiting [23,24]. Consequently, it is necessary to find more novel effective AGIs with less side effects.

1-deoxynojirimycin (1-DNJ), a kind of polyhydroxy alkaloid isolated from some plants of mulberry family, displays a good inhibitory effect on the α-glucosidase, and therefore is regarded as a potential precursor for the synthesis of AGIs [25]. So far, many DNJ derivatives have been reported to have good hypoglycemic activity, such as N-hydroxyethyl-DNJ [26]. Additionally, some modified products of 1-DNJ showed multiple physiological effects, for example, N-alkyl DNJs [27,28] and N-nonyl-DNJ [29,30] served as highly potent pharmacological chaperones for the treatment of Gaucher Curr and Pompe diseases by ‘rescuing’ mutant enzymes. In our previous study, a series of 1-DNJ derivatives were designed and synthesized by the linkage of 1-DNJ and chalcone through an alkyl chain of different carbon number [31,32]. Among these synthesized 1-DNJ derivatives, DC-5 showed the strongest in vitro inhibitory activity on α-glucosidase with a *Ki* of 10 μM, which was significantly better than that of 1-DNJ and acarbose (the most commonly used α-glucosidase inhibitor at present) [33]. Then, we identified the metabolites of DC-5 in blood, its distribution and bioavailability in vivo by ultraperformance liquid chromatography quadrupole time-of-flight tandem mass spectrometry (UPLC-Q-TOF-MS/MS) [34].

In the present study, the potential in vivo anti-diabetic effects of DC-5 were further investigated by the integration detection and analysis of blood glucose concentration, blood biochemical parameters, tissue section and the gut microbiota of the diabetic rats undergoing oral administration of DC-5. Meanwhile, this paper also provided a deep insight into the effects of a 30-day oral administration of DC-5 on PBG, FBG (fasting blood glucose) and the gut microbiota of the diabetic rats.

## 2. Materials and Methods

### 2.1. Chemicals and Materials

DC-5 (HPLC purity ≥ 99%) was designed, synthesized and purified in our laboratory (Figure 1) according to the method of Lin [33]. Streptozotocin (purity ≥ 95%) was bought from Shanghai McLean Biochemical Technology Co., Ltd. (Shanghai, China). Soluble starch (purity ≥ 95%) was obtained from Xilong Chemical Co., Ltd. (Guangdong, China). Glucose test kit was purchased from Shanghai Rongsheng Bio-pharmaceutical Co., Ltd. (Shanghai, China). Microplates were manufactured by Thermo Scientific, Shanghai. (Shanghai, China).

### 2.2. Animals and Diets

The male Sprague Dawley (SD) rats were supplied by Hunan long Shaslek Jingda Laboratory Animal Co., Ltd. (Hunan, China) (license No: scxk 2016-0002). High fat and sugar fodder were also purchased from Hunan long Shaslek Jingda Laboratory Animal Co., Ltd. (Hunan, China), which consisted of 20% sucrose, 10% lard, 2% cholesterol, 1% sodium cholate and 67% basic fodder. The animals were housed in a controlled environment with light/dark cycles of 12/12 h and temperature of 23–25 °C, with free access to food and water. All animal experiments were conducted according to the animal management and experiment guidelines of Jiangxi Agricultural University.

### 2.3. In Vivo Experiments Design

After a week of daily diet and dietary balance adaptation, 96 rats (180–200 g) were randomly divided into two groups including the normal group (*n* = 48) and DM group (*n* = 48).

The normal group was fed with normal diet for 3 weeks, then was randomly divided into the following six groups: control group, acarbose group (10 mg/kg, used as positive drug control group), 1-DNJ Group (10 mg/kg body weight), low DC-5 dosage group (5 mg/kg body weight), medium DC-5 dosage group (10 mg/kg body weight), and high DC-5 dosage group (20 mg/kg body weight). Each group consisted of 8 rats.

The DM group was randomly divided into negative control group (*n* = 8) and model group (*n* = 40). The negative control group was fed with normal diet, while the model group was fed with high fat and sugar diet for 3 weeks. Diabetic model rats were developed according to the method reported by K. Srinivasan et al. [35]. STZ, which was dissolved in a citric acid-sodium citrate buffer solution with a pH of 4.2–4.5, was rapidly injected into abdominal cavity of model group rats after fasting for 16 h, while the negative control group rats were injected with the citric acid-sodium citrate buffer solution of the same volume. Three days after the STZ injection, blood samples were collected from the tail vein for the FBG detection. Rats were then challenged with intragastric administration of starch (2.0 g/kg body weight) for the test of glucose tolerance. If FBG ≥ 11.1 mmol/L and PBG (2 h after meal) ≥ 16.7 mmol/L, the rat was identified as DM rat and used for the further experiments. In subsequent tests, DM rats were randomly divided into the following five groups: model group, acarbose group (10 mg/kg body weight), low DC-5 dosage group (2.5 mg/kg body weight), medium DC-5 dosage group (5 mg/kg body weight), high DC-5 dosage group (10 mg/kg body weight). Each group consisted of 8 rats. Acarbose was used as positive control compound according to the method of Li et al. [36]. At a fixed time of each day, rats were respectively gavaged with DC-5 and acarbose continuously for 30 days.

Rats were fasted for 12 h after a continuous 30-day intragastric administration of DC-5 or acarbose, and then were weighed, killed and dissected successively.

### 2.4. Determination of Body Weight, Food and Water Intake

The body weight of rats was determined at 9 a.m. every day. The amounts of remaining diet and water were also weighted at the same time for the calculation of the average intake of food and water.

### 2.5. Fasting Blood Glucose Test Method

A blood glucose test was performed referring to previously reported methods with some modification [37]. After 12 h of fasting, blood samples were taken from the caudal vein for FBG determination. All blood samples were stored at 4 °C for 0.5 h and then centrifuged at 5000 rpm for 10 min. After centrifugation, the supernatant (plasma) was collected for the blood glucose test. Blood glucose was detected in a microplate reader (Thermo Scientific, Shanghai, China) using a glucose test kit (Shanghai Rongsheng Bio-pharmaceutical Co., Ltd. Shanghai, China), according to the manufacturer’s instructions.

### 2.6. Postprandial Blood Glucose Test Method

According to 2 g/kg body weight, soluble starch was prepared and gelatinized in boiling water bath for 15 min. After cooling, the gelatinized starch was mixed with 1 mL of each sample solution, respectively. As for the negative control group, the sample solution was replaced with 20% anhydrous ethanol for the soluble starch preparation. Blood samples were taken from the caudal vein at 0.5, 1, 1.5, 2 and 3 h after oral administration. The pretreatment of blood samples and the test of blood glucose were conducted in accordance with the method described in Section 2.5.

### 2.7. Visceral Organs Collection and Organ Coefficient Determination

Visceral organs collection and organ coefficient determination was conducted referring to the method of Zheng et al. with some modification [38]. After a continuous 30-day intragastric administration of DC-5, the rats were fasted for 12 h, and then weighed, killed and dissected successively. Blood samples were extracted from the heart for biochemical analysis. The heart, liver, spleen, lung, kidney, stomach and small intestine were taken out, respectively, for further detection. After being washed by physiological saline repeatedly and wiped with filter paper, the visceral organs were weighed for organ coefficient determination. The organ coefficient was calculated by the following formula: Organ coefficient = (W_1_/W_2_) × 100%, where W_1_ and W_2_ were organ weight and body weight, respectively.

### 2.8. Blood/Serum Biochemical Analysis

TG, TC, HDL-C, LDL-C, ALT, AST, CRE, BUN were measured using assay kits (Jiancheng Biological Engineering Institute, Nanjing, China) according to the manufacturer’s instructions.

### 2.9. Preparation and Staining of Sections

The preparation and staining of sections of the collected liver, kidney and small intestine were conducted referring to the method of Jia et al. with some modification [39]. The collected fresh liver, kidney and small intestine were cut into several pieces, and then immersed into 4% polyformaldehyde solution for more than 24 h, respectively. The fixed tissues were embedded in the paraffin and cut into 5 µm sections. After H&E staining, the sections were observed and characterized with an inverted microscope.

### 2.10. Detection and Analysis of Gut Microbiota

#### 2.10.1. Collection of Intestinal Contents

After a continuous 30-day gavage of DC-5 or acarbose, the rats were sacrificed and dissected. The intestinal contents were taken out from the small intestine and stored at −80 °C quickly for DNA extraction, 16S rDNA sequencing, and bioinformatic analysis and statistics.

#### 2.10.2. Extraction and Sequencing of DNA

The DNA of gut microbiota was extracted and sequenced with reference to the previously reported method [40]. The amplification procedure and amplification primers of 16S rDNA gene were shown in Annex 1. The extraction and sequencing of DNA were performed at Biomarker Technologies Co., Ltd. (Beijing, China).

#### 2.10.3. Bioinformatic Analysis and Statistics

Firstly, the generated raw data of gut microbiota sequencing was filtered by Trimmomatic (version 0.33), followed by trimming using Cutadapt (version 1.9.1) to remove primers [41]. Then FLASH [42] (version 1.2.11) and UCHIME [43] (version 8.1) were used to splice the paired-end reads and remove the chimaera. After the joining of paired-end reads and filtering out of low-quality reads, amplicon sequence variants (ASVs) were obtained through DADA2 in QIIME2 (versoin 2020.6) [44]. OTU cluster analysis was accomplished by USEARCH (version 10.0) [45]. The α-diversity analysis software was QIIME2 which was provided by Biomarker Technologies Co., Ltd. (Beijing, China). The β-diversity analysis method based on binary Jaccard, Bray Curtis, (un) weighted UniFrac (bacteria Limited) algorithms was used to appear the species diversity matrix. Bioinformatic analysis and statistics were performed at Biomarker Technologies Co., Ltd. (Beijing, China).

### 2.11. Statistical Analysis

All the determinations were performed at least in triplicate, and the data were expressed as the mean ± standard deviation (SD). Plotting and data analysis were conducted with the software of Origin Pro 8.5.0 (Origin Lab Co., Northampton, MA, USA). Statistical analyses were conducted using SPSS 20.0 for Windows (SPSS Inc., Chicago, IL, USA). One-way analysis of variance was carried out, and means were compared using Duncan’s multiple range tests. Significant differences were determined at *p* < 0.05. The correlation analysis was conducted using a two-tailed test.

## 3. Results

### 3.1. Effects of DC-5 on Postprandial Blood Glucose in Normal Rats

In order to evaluate the effect of DC-5 on the PBG of the normal rats, a total of 48 rats were divided into six groups and treated with different feeding doses, respectively, using acarbose and 1-DNJ (precursor of DC-5) as positive control. As shown in Figure 2A, the PBG (postprandial blood glucose) concentrations of six groups showed a similar change profile, which increased first and then decreased in general, but significantly differed in the peak PBG concentration and time. Peak PBG concentration of each DC-5-treatment group was significantly lower than that of control group, and showed an obvious dose-effect relationship. Although the 1-DNJ group displayed a noteworthy inhibitory effect on the PBG, but its peak concentration was still higher than that of the low DC-5 dose group. The high dose group showed an excellent inhibition effect on the PBG, whose PBG concentration was even lower that of acarbose, which was the most commonly used α-glucosidase inhibitor at present. Except for 1-DNJ giving a PBG peak time at 90 min after gavage, all of other groups peaked at 60 min. The PBG of all groups decreased from postprandial 90 min. Both high dose group and acarbose group showed a low PBG at 180 min after gavage, which almost decreased to the level of FBG (fasting blood glucose). As shown in Figure 2B, the AUC (area under the curve) of all treatment groups were significantly lower than that of control group, especially the high DC-5 dose group and acarbose group. Meanwhile, the AUC of all DC-5 treatment groups was markedly lower than that of the 1-DNJ group, and showed dose-response relationship (Figure 2B). The above results indicated that DC-5 had an inhibitory effect on the PBG in normal rats.

### 3.2. Effect of DC-5 on FBG and PBG in Diabetic Rats

Based on our above-mentioned in vivo experiments on the effects of DC-5 on postprandial blood glucose in normal rats, we conducted pre-experiments for the in vivo hypoglycemic effect evaluation with diabetic model rats, using a set of DC-5 doses of 2.5, 5, 10, 20 mg/kg body weight. The results suggested that DC-5 at a dose of 2.5 to 10 mg/kg body weight has already shown an obvious hypoglycemic effect. Consequently, DC-5 doses of 2.5, 5, 10 mg/kg body weight were used to evaluate the hypoglycemic effect of DC-5 in diabetic model rats.

#### 3.2.1. Effects of DC-5 on FBG in Diabetic Rats

As shown in Table 1, the FBG of the control group was maintained between 6.17–6.79 mmol/L during the whole experiment, while the FBG of the other groups was all higher than 11.1 mmol/L and no significant difference was observed between groups at the beginning of the gavage experiment, which indicated that diabetic rats were successfully induced and our experiment grouping is reasonable. Overall, the FBGs of all treatment groups and the model group increased at first and then decreased from the beginning to the end of the experiment, and peaked on the 20th or 15th day (Table 1). Compared with acarbose, DC-5 showed a better reduction effect on the FBG. The FBG of low, medium and high DC-5 dose groups decreased to 22.28, 19.14 and 16.16 mmol/L, respectively, on the 30th day of intragastric administration experiment, and presented an obvious dose-effect relationship. The above-mentioned result and analysis suggested that DC-5 had a remarkable improvement effect on the FBG of diabetic rats.

#### 3.2.2. Effects of DC-5 on PBG in Diabetic Rats

To evaluate the in vivo inhibitory effects of DC-5, the PBG of diabetic rats were detected on the 5th and 30th day of intragastric administration experiment (Figure 3A,B). On the whole, the PBGs of all DC-5 treatment groups were very significantly higher than that of the negative control group (normal rats), but obviously lower than that of the model group (untreated diabetic rats). In general, the high DC-5 dose group revealed the best inhibitory effect, followed by acarbose, medium and low dose group, successively. Although the inhibitory effect of the low DC-5 dose was relatively weak, an obvious reduction effect of the PBG could still be observed in this group. With regard to the PBG peak time, all peaks of the treated diabetic rats (including acarbose treated rats) appeared at the 30th min, except the high DC-5 dose group on the 5th day of the gavage experiment, while the peak time was postponed to the 90th min on the 30th day. As shown in Figure 3C, the AUC (area under curve) of all treated groups was significantly higher than that of the control group (1202.1 mg/mL·min), but lower than that of the untreated diabetic rats. The lowest AUC was observed in the high DC-5 dose group. The AUC of acarbose group was similar to that of the low DC-5 group. The above analysis suggested that a 30-day DC-5 treatment effectively improved the PBG of diabetic rats.

### 3.3. Effects of DC-5 on Body Weight, Water Intake and Food Intake of Diabetic Rats

#### 3.3.1. Effects of DC-5 on Body Weight in Diabetic Rats

As shown in Figure 3D, the body weight of the control group increased linearly until the 27th day, while the body weight of the untreated DM model, the DC-5 and the acarbose treated groups presented different change profiles during the whole intragastric administration experiment. In the first six days, a small body weight increment was seen in all of the untreated DM models, the DC-5 and acarbose treated groups, but different change trends emerged after the 6th day. The body weight of the DM model rats began to decrease on the 12th day, and kept a slow downward trend until the 30th day, while the body weight of the high DC-5 dose group basically remained unchanged. However, unlike the DM model and the high DC-5 dose groups, the body weight of the acarbose, low and medium DC-5 dose groups generally showed a slow upward trend, especially the acarbose-treated rats, whose body weigh were significantly higher than that of other treatment groups since the 15th day. The above results indicated that DC-5 could ameliorate the body weight loss of diabetic rats.

#### 3.3.2. Effects of DC-5 on Water Intake of Diabetic Rats

The water intake amounts of the treated and untreated diabetic rats were all extremely significantly greater than that of the normal rats, which were kept at around 50 g/day (Figure 3E). Throughout the trial process, rats of the DM model group generally showed the highest water intake amount, followed by the acarbose, low, medium and high DC-5 dose groups. It should be noted that the water intake amounts of the high DC-5 dose group were always obviously lower than that of other treatment groups; the differences could be clearly seen since the 9th day. Summarily, DC-5 displayed an obvious improvement effect on the polydipsia of diabetic rats.

#### 3.3.3. Effects of DC-5 on Food Intake of Diabetic Rats

Similar to the effects on water intake, DC-5 demonstrated a positive effect on the polyphagia of diabetic rats. At the beginning of the experiment, the food intake amounts of the treated and untreated diabetic rats all increased sharply, and then kept a relatively slower upward tendency, while the food intake amounts of the normal rats presented a much slower upward trend from beginning to end (Figure 3F). Likewise, the high DC-5 dose group (10 mg/kg body weight) showed the best improvement effect on the polyphagia of diabetic rats.

### 3.4. Effects of DC-5 on Blood Biochemical Indexes of Diabetic Rats

#### 3.4.1. DC-5 Improved Serum Lipid Profiles of Diabetic Rats

It could be seen from Figure 4A,B that TG, TC and LDL-C of the DM model rats were remarkably higher than that of the normal rats, while HDL-C was significantly lower that of the normal rats, indicating that DM rats had an obvious disorder symptom in blood lipid metabolism. Compared with the model group, the TG, TC and LDL-C of diabetic rats all decreased after a 30-day treatment of DC-5 and acarbose, meanwhile the HDL-C increased conversely, appearing a dose-effect relationship, which suggested that DC-5 could repair the disorder of lipid metabolism of diabetic rats. High dose DC-5 displayed the best recovery effect, whose TC, TG, LDL-C and HDL-C were 8.98 mmol/L, 1.08 mmol/L, 5.01 mmol/L and 4.01 mmol/L, respectively, at the 30th day. This experiment indicated that DC-5 had a fine recovery effect on the lipid metabolism disorder of diabetic rats, even better than acarbose.

#### 3.4.2. Protective Effects of DC-5 on Liver and Kidney of Diabetic Rats

ALT and AST are commonly used to evaluate the liver function of animals. As shown in Figure 4E,F, acarbose and DC-5 treatment (including low, medium and high dose) lowered the elevated ALT and AST of the diabetic rats. In comparison, the high DC-5 dose displayed the best effect. The ALT and AST were 5.46 U/L and 24.17 U/L after the 30-day treatment of high dose DC-5.

BUN and CRE are terminal products of protein metabolism, and usually regarded as key indicators of kidney function. As shown in Figure 4C,D, acarbose and DC-5 treatment (including low, medium and high dose) significantly reduced the elevated BUN and CRE of the diabetic rats. Overall, there were not much difference in BUN and CRE among the low, medium and high dose group, but the recovery effect of DC-5 was obviously better than acarbose. The impaired BUN and CRE decreased from 29.08 mmol/L and 81.04 mmol/L to 10.88 mmol/L and 34.55 mmol/L after the 30-day treatment of high dose DC-5, which was significantly better than that of acarbose treatment.

As shown in Table 2, the organ coefficient of the DM model group was significantly higher than that of the control group. After the 30-day treatment of DC-5 and acarbose, the raised coefficient of liver (5.29%) and kidney (1.31%) declined obviously, indicating that DC-5 had an improvement effect on the elevated organ coefficient of diabetic rats.

### 3.5. Histological Analysis of Diabetic Rats Treated by DC-5

#### 3.5.1. Analysis of Liver Tissue Sections

From Figure 5A, it could be seen that the liver cells of the normal rats (control group) were arranged neatly and spaced regularly with complete structure and clear boundary, while the cells of the diabetic rats (DM model group) shrank and were accompanied by severe hepatic lipid accumulation and cytoplasmic vacuoles. After a 30-day treatment of DC-5, the morphology of the impaired liver cells of diabetic rats was improved obviously (Figure 5A). Relatively speaking, the low and medium DC-5 dose showed a better recovery effect. After a month of oral administration of the low and medium dose DC-5, the pathological changes were reversed; meanwhile the atrophy and fat vacuole almost disappeared, and the impaired liver cells even returned to the state of normal cells.

#### 3.5.2. Analysis of Kidney Tissue Sections

As shown in Figure 5B, there was no obvious difference between the normal and diabetic rats in the kidney overall structure and the morphology of glomerulus and lumen. In the meantime, no obvious change was observed in the kidney tissue sections of the diabetic rats, which underwent a 30-day treatment of DC-5 (including low, medium and high dose), suggesting that DC-5 of dosage no more than 10 mg/kg had no adverse effect on the kidney of rats.

#### 3.5.3. Analysis of Small Intestinal Tissue Sections

Figure 5C displayed the small intestinal tissue sections of the normal rats, diabetic rats and diabetic rats treated by low, medium and high dose DC-5, respectively. The small intestinal tissue of the normal rats presented a complete structure with long villus and finger-like protrusions, and the circular muscle, mucosal muscle layer and abundant small blood vessels were clearly visible, while the villi structure of the diabetic rats was impaired to some extent, and the villus looked incomplete and a little vague. After a month of oral administration of DC-5, pathological changes of the intestinal tissue of diabetic rats were markedly reversed. As shown in Figure 5C, the impaired villi structure almost fully recovered with complete and clear finger-like protrusions after a 30-day treatment of medium and high dose DC-5.

### 3.6. Overall Structural Modulation of Gut Microbiota in Diabetic Rats Treated by DC-5

#### 3.6.1. Analysis of Population Diversity

The results of 16S rDNA sequencing (Figure 6) indicated that a total of 1205 OTU were identified (Figure 6A), and the OTU numbers of the normal rats, DM model rats and diabetic rats treated by high dose DC-5 were 1123, 908 and 1057, respectively (Figure 6B). A total of 832 OTU was found in all the above-mentioned three groups, and the unique OTU numbers of the three groups were 108, 2 and 44, respectively (Figure 6B). From the test data of Figure 6, it could be seen that the intestinal flora diversity of the DM model rats reduced obviously compared with that of the normal rats, but declined gut microbiota abundance had been completely recovered by the DC-5 treatment. After a 30-day oral administration of DC-5 (10 mg/kg body weight), the OTU number of the diabetic rats increased by 43.32% (from 2253 to 3229), which even exceeded the number of the normal rats.

#### 3.6.2. Alpha Diversity Analysis

The results of alpha diversity analysis (Table 3) showed that the Shannon and Chao1 index of the DM model rats decreased significantly from 6.35 and 798, to 43.9 and 687, but the reduced indexes had been basically restored under the DC-5 treatment. Through a month of treatment, DC-5 (10 mg/kg body weight) significantly improved the decreased intestinal flora diversity of the diabetic rats, and the ACE, Chao1, Simpson and Shannon indexes rose by 29.55%, 28.68%, 3.45% and 30.52%, respectively.

#### 3.6.3. Beta Diversity Analysis

The PCA analysis (as shown in Appendix A) suggested that the normal rats (group A in Appendix A) exhibited a high similarity, and therefore clustered commendably. On the contrary, the DM model rats (group B in Appendix A) scattered over a relatively larger area, and obviously separated from group A. After a 30-day oral administration of DC-5 (10 mg/kg body weight), although the treated diabetic rats (group C in Appendix A) also displayed a certain degree of discreteness, the scatter points evidently separated from group B and located closer to group A, which indicated that the impaired intestinal flora diversity had been recovered to a certain extent.

### 3.7. Community Composition Change Analysis of Gut Microbiota in Diabetic Rats Treated by DC-5

The data of 16S rDNA sequencing were further analyzed at the phylum level. All of the identified bacteria were divided into 12 categories (phylum), such as *Firmicutes*, *Proteobacteria, Bacteroidetes,* etc., which were shown in Figure 7. On the whole, all the top four phyla of the normal rats, the DM model rats and the diabetic rats treated by high dose DC-5 were *Firmicutes*, *Proteobacteria, Bacteroidetes* and *Cyanobacteria*. According to the relative abundance, the principal bacteria of the normal rats was *Firmicutes*, and the second and third were *Proteobacteria* and *Cyanobacteria,* respectively. While in the intestinal tract of the DM model rats, *Firmicutes* and *Bacteroidetes* increased markedly, while *Proteobacteria* and *Cyanobacteria* decreased significantly, suggesting that gut microbiota in the diabetic rats changed greatly at the phylum level. After a 30-day treatment of high dose DC-5 (10 mg/kg body weight), the composition of intestinal flora had been well recovered, the reduced percentages of *Proteobacteria* and *Cyanobacteria* ascended significantly again, and the declined ratio of *Firmicutes* to *Bacteroidetes* also raised obviously once more. The above results and analysis suggested that DC-5 was beneficial to the restoration of intestinal flora, which might further promote the recovery of the diabetic rats.

## 4. Discussion

In recent years, the world economy has developed rapidly, which results in great changes of people’s living style, and further increases the incidence rate of T2D (type 2 diabetes) and its complications, all over the world. Consequently, it is urgent to develop effective pharmaceuticals for the treatment of T2D and its complications [46]. α-Glucosidase inhibitors can effectively reduce postprandial hyperglycemia [47], and therefore are regarded as a category of preferred oral hypoglycemic medicines and are widely used in clinic at present. In a previous study [33], a series of novel N-alkyl-1-deoxynojirimycin derivatives were synthesized to develop α-glucosidase inhibitors with high activity, and obtained 20 compounds, which exhibited in vitro inhibitory activity on α-glucosidase with IC_50_ values ranging from 30 to 2000 mM, among which DC-5 was the most effective compound with a IC_50_ of 10 mM, and showed a promising application prospect in the treatment of diabetes mellitus. However, the in vivo anti-diabetes activity of this compound has not yet been verified up to now. In the present study, the potential in vivo anti-diabetic effects of DC-5 were further investigated by the integration detection and analysis of blood glucose concentration, blood biochemical parameters, tissue section and the gut microbiota of the diabetic rats. Our results suggested that DC-5 had a good in vivo hypoglycemic effect in the diabetic rats, improved diabetes symptoms, and alleviated the pathological changes of the liver, kidney and small intestine of the diabetic rats, and meanwhile promoted the recovery of the gut microbiota of diabetic rats.

1-deoxynojirimycin (1-DNJ) is a polyhydroxylated alkaloid, which was firstly identified from the mulberry tree. This compound displays an effective inhibitory activity on various carbohydrate-degrading enzymes, which were involved in a wide range of important biological processes [48]. It was reported that the treatment of a mixture of deoxynojirimycin and polysaccharide (150 mg/kg body weight) for 90 days significantly decreased the blood glucose, pyruvate, TG, AST, ALT, Cr, lipid peroxide and malondialdehyde levels [49]. In other research, 1-DNJ was found to have a gender-specific modulating effect on hypercholesteremia and gut microbiota [50]. However, there were still some reports with different conclusion in regard to the in vivo hypoglycemic effect of 1-DNJ. According to the studies of Nakagawa et al. and Faber et al., 1-DNJ and DMJ (with a similar structure of 1-DNJ) were rapidly excreted with the urine in an intact form, which resulted in a weak in vivo effect on reducing blood glucose [51,52]. Consequently, 1-DNJ modification attracted the attention of researchers, and some novel derivatives were synthesized successively with a high inhibitory activity on α-glucosidase, for example, Quinazoline-1-deoxynojirimycin hybrids [53], N-alkyl-deoxynojirimycin [54] and DNJ neoglycoconjugates [55]. In our laboratory, DC-5 was synthesized using 1-DNJ as a precursor and showed a much better in vitro α-glucosidase inhibitory activity, better than 1-DNJ [33]. In the present paper, DC-5 was again confirmed to have an outstanding in vivo anti-diabetic effect. Therefore, this 1-DNJ derivatives is a high potential anti-diabetic pharmaceutical.

Researchers had demonstrated that gut microbiota was closely related to the occurrence of metabolic diseases such as diabetes and obesity, and the change of gut microbiota might be one of the causes of diseases related to metabolic syndrome [56]. A recent study indicated that the relative abundance of *Bacteroidetes* and the ratio of *Bacteroidetes*/*Firmicutes* significantly increased in T2D rats [57]. Another investigation also showed that 1-DNJ obviously alleviated the gut microbiota imbalance in diabetic mice [58]. Our test results suggested that DC-5, a novel derivative of 1-DNJ, not only displayed an outstanding in vitro and in vivo hypoglycemic effect, but also markedly improved the ratio of *Firmicutes* to *Bacteroides,* and well restored the impaired intestinal flora, which might be one of the important mechanisms to promote the recovery of diabetes.

## 5. Conclusions

DC-5 reduced the PBG and the AUC of normal rats, and effectively decreased the impaired PBG and the FBG of diabetic rats. Furthermore, DC-5 lowered the levels of TG, TC, LDL-C, BUN and CRE, and elevated the level of HDL-C; meanwhile, it alleviated the adverse lesions of the liver, kidney and small intestine of diabetic rats. Moreover, DC-5 could repair gut microbiota disorder, and improve the ratio of *Firmicutes* to *Bacteroides,* which might further promote the recovery of diabetic rats. In conclusion, DC-5 displayed a positive in vivo treatment effect on diabetes mellitus, and therefore has a potential application prospect.

## Figures and Tables

**Figure 1 molecules-27-07583-f001:**
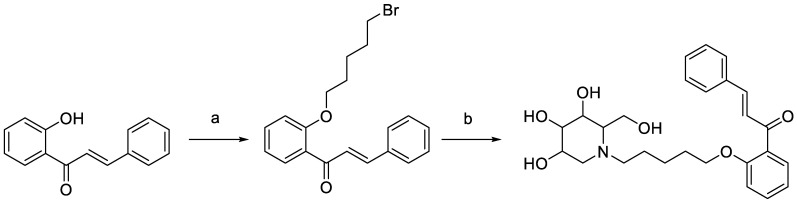
Synthesis scheme of intermediate and target products of DC-5. Reagents and conditions: (**a**) K_2_CO_3_, acetone, dibromopentane, 65 °C, overnight; (**b**) 1-DNJ, K_2_CO_3_, DMF, 85 °C, 6 h.

**Figure 2 molecules-27-07583-f002:**
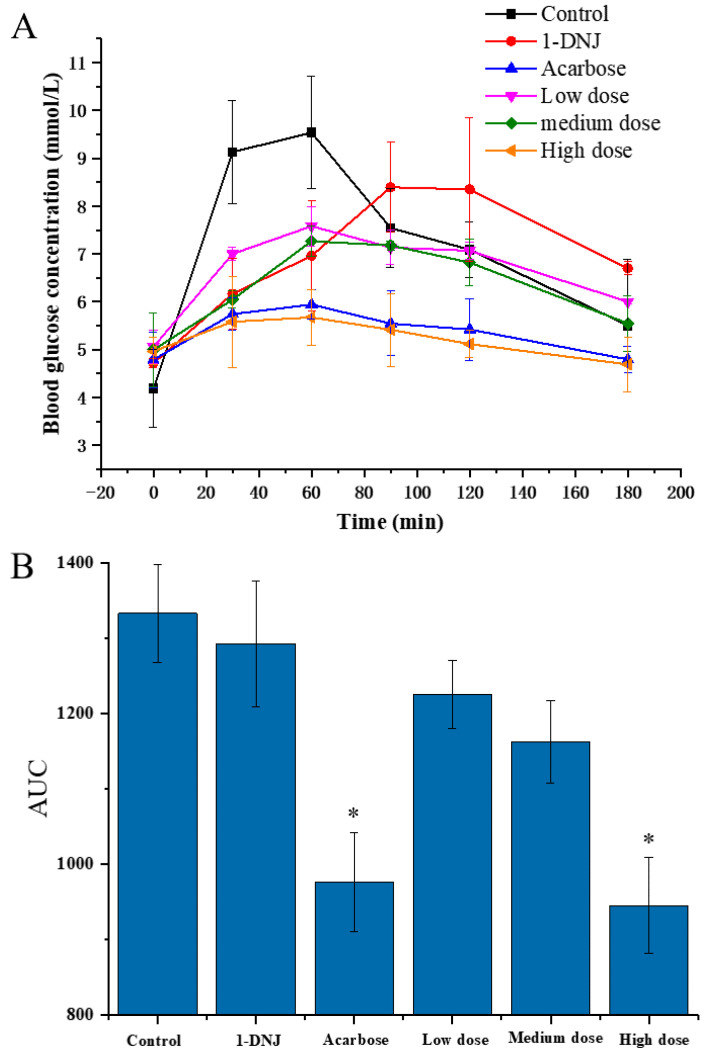
Postprandial blood glucose (**A**) and area under the curve of PBG (**B**) of normal rats treated or untreated by DC-5 at different doses. Where * indicates *p* ≤ 0.05 when compared with control group.

**Figure 3 molecules-27-07583-f003:**
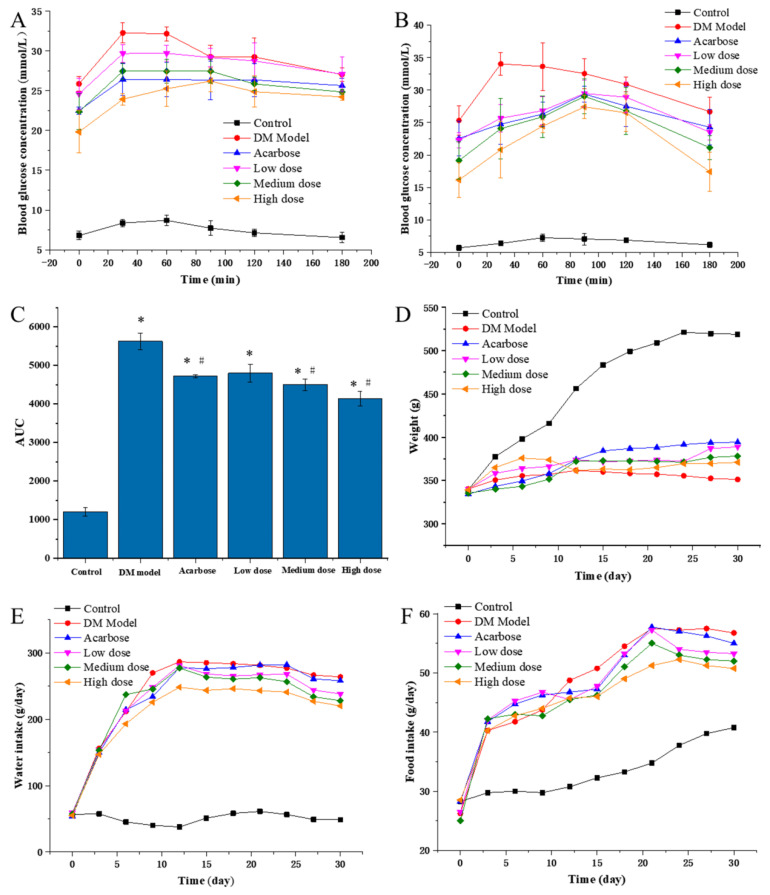
PBG ((**A**) PBG on the 5th day of treatment; (**B**) PBG on the 30th day of treatment), AUC (**C**), body weight (**D**), water and food intake (**E**,**F**) of the normal rats, diabetic rats and diabetic rats treated by acarbose or DC-5 at different doses for 30 days. Where * indicates *p* ≤ 0.05 when compared with control group; # indicates *p* ≤ 0.05 when compared with model group.

**Figure 4 molecules-27-07583-f004:**
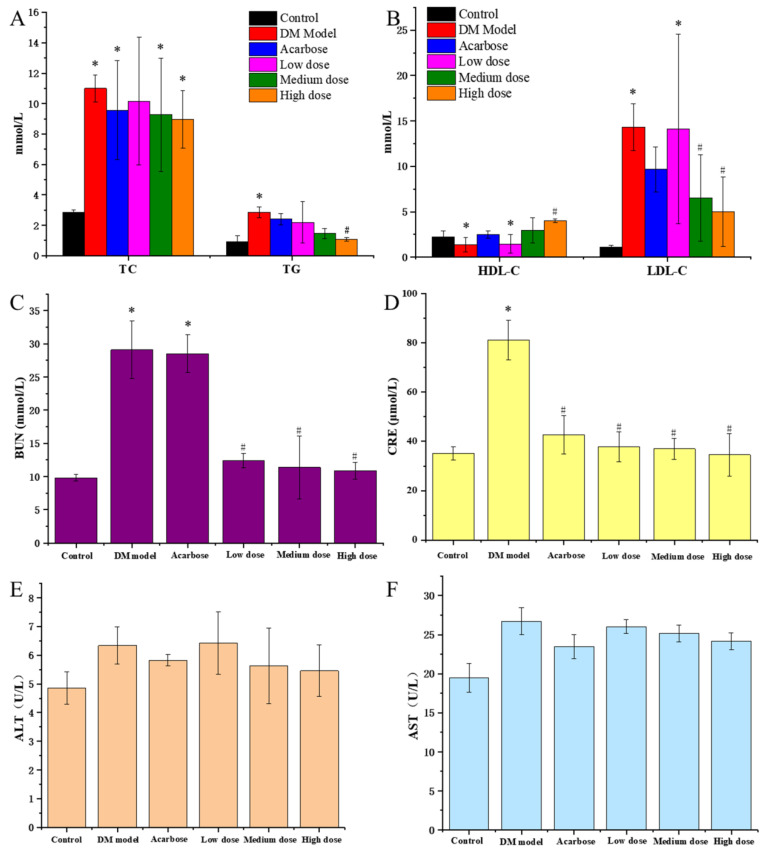
TC and TG (**A**), HDL-C and LDL-C (**B**), BUN (**C**), CRE (**D**), ALT (**E**) and AST (**F**) of the normal rats, untreated diabetic rats and diabetic rats treated by acarbose or DC-5 at different doses for 30 days. All the indicators were detected on the 30th day of the gavage experiment. Where * indicates *p* ≤ 0.05 when compared with control group; # indicates *p* ≤ 0.05 when compared with model group.

**Figure 5 molecules-27-07583-f005:**
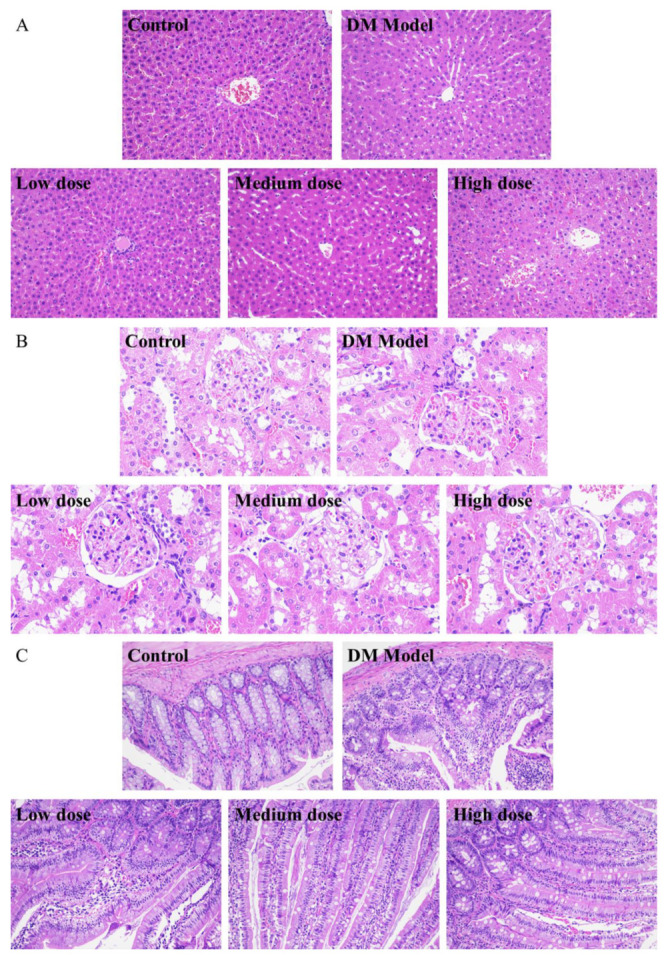
H&E staining sections of liver (**A**), kidney (**B**), small intestine (**C**) of the normal rats, untreated diabetic rats and diabetic rats treated by DC-5 at different doses for 30 days. All the samples for section preparation were collected on the 30th day of the gavage experiment. (**A**,**C**) were observed at 200× magnification. (**B**) was observed at 400× magnification.

**Figure 6 molecules-27-07583-f006:**
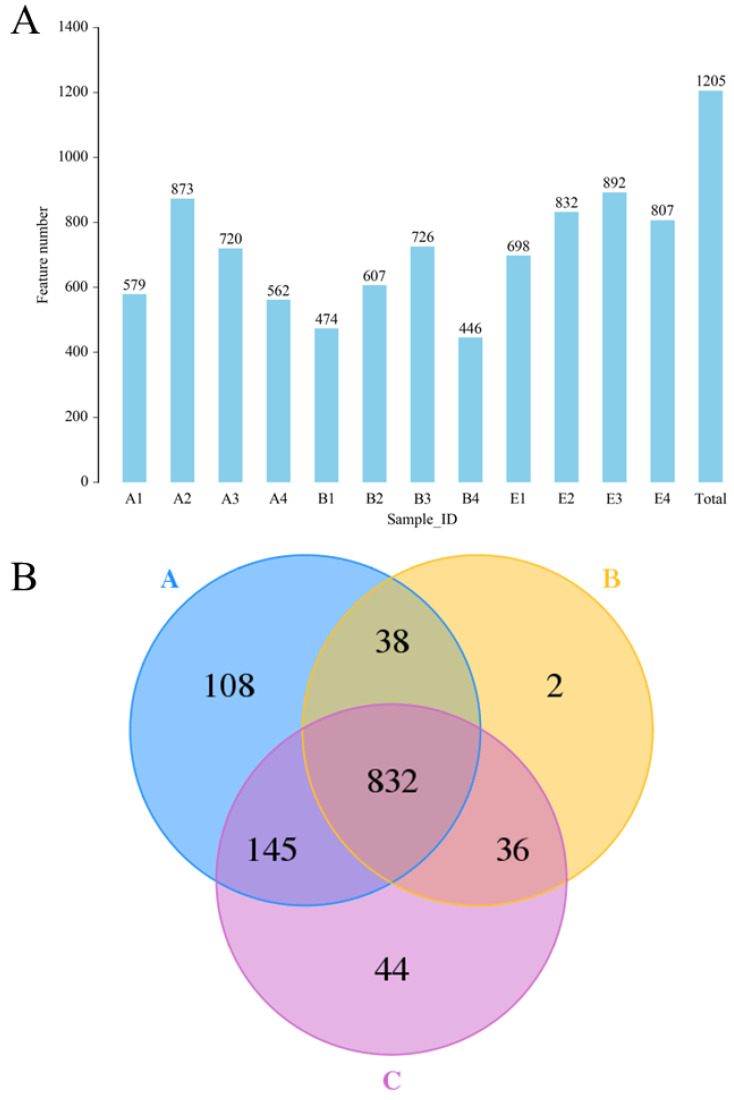
The identified OTU numbers of the normal rats, untreated diabetic rats and diabetic rats treated by high dose DC-5 (10 mg/kg body weight) for 30 days (**A**) and Venn diagram of the identified OTU (**B**). OTUs were identified by 16S rDNA sequencing. Where, A1–A4: normal rats; B1–B4: untreated diabetic rats; E1–E4: diabetic rats treated by high dose DC-5. A: normal rat group; B: untreated diabetic rat group; C: diabetic rat group treated by high dose DC-5.

**Figure 7 molecules-27-07583-f007:**
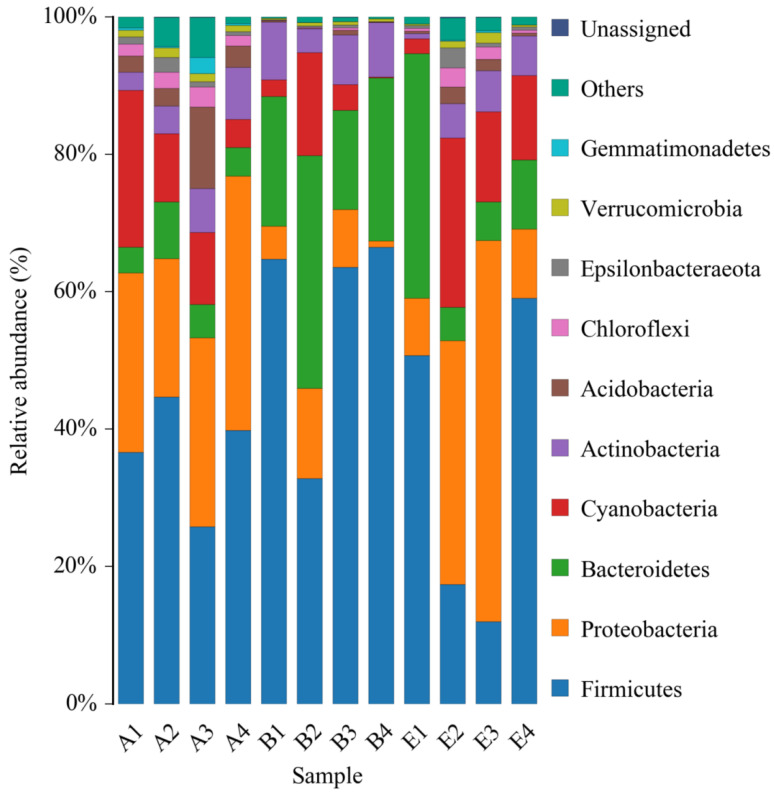
Relative abundance of bacterial phyla of the normal rats, untreated diabetic rats and diabetic rats treated by high dose DC-5 (10 mg/kg body weight) for 30 days. Where, A1–A4: normal rats; B1–B4: untreated diabetic rats; E1–E4: diabetic rats treated by high dose DC-5.

**Table 1 molecules-27-07583-t001:** Summary sheet of FBG of diabetic rats treated by DC-5 and acarbose for 30 days (Mean ± SD, mmol/L).

Gavage Day	Control	DM Model	Acarbose	Low Dose	Medium Dose	High Dose
1st day	6.33 ± 0.68	14.95 ± 1.61	15.06 ± 3.50	15.99 ± 3.44	15.83 ± 2.54	14.25 ± 2.23
5th days	6.79 ± 0.53	25.86 ± 0.89	22.50 ± 0.47	24.60 ± 1.93	22.37 ± 2.23	19.82 ± 2.66
15th days	6.11 ± 0.76	27.81 ± 2.72	26.45 ± 2.30	24.78 ± 0.64	24.02 ± 2.10	21.95 ± 3.07
20th days	6.28 ± 0.87	28.44 ± 0.41	28.14 ± 1.68	24.65 ± 1.71	24.30 ± 2.86	23.33 ± 1.68
25th days	6.23 ± 0.60	26.26 ± 2.42	27.16 ± 2.80	20.44 ± 3.58	20.46 ± 2.68	20.93 ± 4.22
30th days	6.17 ± 0.54	26.66 ± 2.06	24.27 ± 2.78	22.28 ± 1.19	19.14 ± 2.89	16.16 ± 2.72

Note: DC-5 dosage: Low dose: 2.5 mg/kg body weight; medium dose: 5 mg/kg body weight; high dose: 10 mg/kg body weight. Acarbose dosage: 10 mg/kg body weight.

**Table 2 molecules-27-07583-t002:** Summary sheet of organ coefficient of diabetic rats treated by DC-5 and acarbose for 30 days (Mean ± SD).

Organ Name	Control	DM Model	Acarbose	Low Dose	Medium Dose	High Dose
Liver	3.02 ± 0.13	5.29 ± 0.07	4.35 ± 0.3	4.29 ± 0.37	4.59 ± 0.38	4.80 ± 0.23
Kidney	0.68 ± 0.07	1.31 ± 0.08	1.09 ± 0.06	1.19 ± 0.13	1.15 ± 0.03	1.26 ± 0.14

Note: DC-5 dosage: Low dose: 2.5 mg/kg body weight; medium dose: 5 mg/kg body weight; high dose: 10 mg/kg body weight. Acarbose dosage: 10 mg/kg body weight.

**Table 3 molecules-27-07583-t003:** Comparison table of Alpha diversity indexes between normal rats, diabetic rats and DC-5 treated diabetic rats.

	Feature	ACE	Chao1	Simpson	Shannon	Coverage
Control	684 ± 145	811 ± 119	798 ± 99	0.93 ± 0.04	6.35 ± 0.99	0.9987 ± 0.0005
DM Model	563 ± 129	670 ± 101	687 ± 93	0.87 ± 0.04	4.39 ± 0.66 *	0.9983 ± 0.0006
DC-5	807 ± 81	868 ± 46 #	884 ± 37 #	0.90 ± 0.05	5.73 ± 0.79 #	0.9985 ± 0.0006

Note: * represents *p* ≤ 0.05, *t* test, compared with control group rats (normal rats); # represents *p* ≤ 0.05, *t* test, compared with model group rats (diabetic rats). Dosage of DC-5: 10 mg/kg body weight.

## Data Availability

Data are contained within the manuscript.

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
