# Peer review of "Chalcone-1-Deoxynojirimycin Heterozygote Reduced the Blood Glucose Concentration and Alleviated the Adverse Symptoms and Intestinal Flora Disorder of Diabetes Mellitus Rats"

_molecules, 2022, doi:10.3390/molecules27217583_

Round 1
Reviewer 1 Report
Xiao et al. submitted the manuscript, "Chalcone-1-deoxynojirimycin heterozygote reduced the blood glucose concentration and alleviated the adverse symptoms and intestinal flora disorder of diabetes mellitus rats" is about the investigation of 1-DNJ tethered with non-substituted diphenyl propanone through a hydrocarbon chain. This synthesized compound (DC-5) was investigated for blood glucose activity.
The biological work is commendable, and the data seems promising. However, the chemistry of the paper needs to revise.
1. Please indicate the reaction between chalcone, 1-DNJ as a synthetic scheme.
2. These precursors are quite common, and 1-DNJ has already been well explored for substitution and should be included in the introduction section of the manuscript.
For more information:
(N-hydroxyethyl-DNJ (glyset®) is used for type-II diabetes-associated complications (Diabet. Med., 15 (1998), pp. 657-660). Also, N-alkyl DNJs (Bioorgan. Med. Chem., 14 (2006), pp. 7736-7744; Bioorg. Med. Chem. Lett., 14 (2004), pp. 5991-5995) such as zavesca®, N-nonyl-DNJ act as highly potent pharmacological chaperones for the potential treatment of Gaucher Curr. Opin. Chem. Biol., 11 (2007), pp. 412-418 and Pompe (Mol. Ther., 17 (2009), pp. 964-971) diseases by ‘rescuing’ mutant enzymes.
3. Because of such therapeutic application, as mentioned in the previous comment-2, N-alkylation is commonly reported with DNJ, therefore, please include suitable references.
4. To confirm the synthesis, usually NMR or other spectroscopy techniques are commonly employed, therefore, incorporate such information to consolidate the synthetic part of the manuscript.
The manuscript contains all the medicinal chemistry elements, but the authors need to revise the above points to consider in the journal.
Reviewer 2 Report
The search for biologically active compounds, especially when molecules could reveal multi-targeted drug properties, is the actual field of research. In this manuscript, the authors chose a compound chalcone-1-deoxynojirimycin heterozygote (DC-5) from a series of 1-DNJ derivatives that were designed and synthesized by the linkage of 1-DNJ and chalcone in their previous studies. In previous studies, the chosen compound showed potent in vitro inhibitory activity on α-glucosidase for diabetes treatment. In the current study, in vivo anti-diabetic studies of DC-5 were carried out. The oral treatment with DC-5 decreases fasting blood sugar and postprandial blood sugar in diabetic and normal rats. While also reducing the negative symptoms of raised blood lipid levels and considerably correcting the gut microbiota disorder in diabetic rats. I find the developed procedure is a valuable method. Overall the results and methods are well explained and concluded.
· However, there are some doubts about the study's rationale and why only one compound was explicitly targeted for in vivo studies.
· The synthesis scheme must be incorporated in the manuscript.
· The compound must be evaluated for cytotoxic studies.
· There are a lot of grammatical, typing, and spelling mistakes which need to be corrected few are highlighted in the attached file.
· In line number 111, kindly write the full form of STZ.

Round 2
Reviewer 2 Report
The authors have improved the manuscript, and now it may be published.